# Anti-Inflammatory Activity of 1,4-Naphthoquinones Blocking P2X7 Purinergic Receptors in RAW 264.7 Macrophage Cells

**DOI:** 10.3390/toxins15010047

**Published:** 2023-01-05

**Authors:** Sergei A. Kozlovskiy, Evgeny A. Pislyagin, Ekaterina S. Menchinskaya, Ekaterina A. Chingizova, Yuriy E. Sabutski, Sergey G. Polonik, Galina N. Likhatskaya, Dmitry L. Aminin

**Affiliations:** G.B. Elyakov Pacific Institute of Bioorganic Chemistry, Far-Eastern Branch of the Russian Academy of Sciences, Vladivostok 690022, Russia

**Keywords:** 1,4-naphthoquinones, Ca^2+^ influx, YO-PRO-1 uptake, P2X7R, macrophages, ROS production, IL-1β and TNF-α release, COX-2 activity, antioxidant activity, inflammation

## Abstract

P2X7 receptors are ligand-gated ion channels activated by ATP and play a significant role in cellular immunity. These receptors are considered as a potential therapeutic target for the treatment of multiple inflammatory diseases. In the present work, using spectrofluorimetry, spectrophotometry, Western blotting and ELISA approaches, the ability of 1,4-naphthoquinone thioglucoside derivatives, compounds **U-286** and **U-548**, to inhibit inflammation induced by ATP/LPS in RAW 264.7 cells via P2X7 receptors was demonstrated. It has been established that the selected compounds were able to inhibit ATP-induced calcium influx and the production of reactive oxygen species, and they also exhibited pronounced antioxidant activity in mouse brain homogenate. In addition, compounds **U-286** and **U-548** decreased the LPS-induced activity of the COX-2 enzyme, the release of pro-inflammatory cytokines TNF-α and IL-1β in RAW 264.7 cells, and significantly protected macrophage cells against the toxic effects of ATP and LPS. This study highlights the use of 1,4-naphthoquinones as promising purinergic P2X7 receptor antagonists with anti-inflammatory activity. Based on the data obtained, studied synthetic 1,4-NQs can be considered as potential scaffolds for the development of new anti-inflammatory and analgesic drugs.

## 1. Introduction

ATP is an important intercellular messenger and neurotransmitter that activates ligand-gated ion channels of the P2X family (P2XR) and metabotropic receptors of the P2Y family [1]. These receptors play an important role in nerve signal transmission, induction of inflammation reactions and cell death [2]. It is also known that an excessive increase in the concentration of ATP in the extracellular space due to direct cell destruction or infection affects stress receptors, such as the P2X7 receptor (P2X7R) [3]. P2XRs are also considered as a part of the innate immune response. In the event of infection, pathogen-associated molecular patterns (PAMPs), in turn, activate pathogen recognition receptors (PRRs) (i.e., Toll-like receptors), which induces ATP release and thus influences P2XR activation [4].

P2X7R is expressed by hematopoietic immune cells, including dendritic and mast cells, T-cells, macrophages, and microglia [5]. Long-term exposure to high concentrations of ATP (>1 mM) on the P2X7 receptor can lead to the opening of a macropore permeable to charged hydrophilic molecules up to 900 Da in size, which can cause necrotic or apoptotic cell death, depending on the tissue type [6]. 

So far, many non-selective P2X7R blockers have been discovered, such as BBG, PPADS, TNP-ATP, and others [7]. Many of them were studied in various pathological conditions and turned out to be active in the early stages of trials. However, the low affinity of these antagonists for these receptors led to low efficiency and required increased dosages; as a result, many of them did not pass preclinical or clinical trials [8]. Currently, the results of pharmacological studies confirm the relevance of the search for new classes of effective and selective P2X7R blockers to understand the fundamental relationships in the functioning of the receptor and to develop new drugs based on them [9].

Inflammation is one of the major problems associated with diseases caused by P2X7R overactivation. However, there are no anti-inflammatory drugs on the pharmaceutical market based on selective blockers of these receptors [8]. Modern anti-inflammatory drugs are divided into two categories: steroidal and non-steroidal anti-inflammatory drugs (NSAIDs). NSAIDs target the activity of the COX-1/2 enzyme [10]. The activity of this enzyme is increased under the action of the pro-inflammatory cytokine IL-1β, the maturation and release of which is directly related to the activation of P2X7R. At the moment, many anti-inflammatory pharmacological drugs are undergoing clinical trials due to the increase in the incidence of inflammatory diseases and the insufficient effectiveness of conventional NSAIDs (ibuprofen, naproxen, diclofenac and many others). Data from numerous placebo-controlled studies and meta-analyses of studies indicate with concern the side effects of NSAIDs in gastrointestinal, hepatic, cardiovascular, renal, cerebral and pulmonary complications [11]. The complexity and diversity of the human inflammatory system requires the search and identification of new molecular targets for the development of drugs for the treatment of inflammatory and autoimmune diseases. Such drugs should have sufficient specificity, minimal side effects and high efficacy. Both natural and synthetically obtained compounds can serve as a source for new drugs [12].

Natural 1,4-naphthoquinones (1,4-NQs) are a promising class of biologically active compounds that perform various protective functions in plant and animal organisms [13]. Most of 1,4-NQs are cytotoxic for different type cells but at low concentrations some of them exhibit pronounced antioxidant and cardioprotective properties in postinfarction conditions and burn conditions associated with extensive tissue damage [14]. Based on natural 1,4-NQs, various synthetic compounds with high toxicity to cancer cells have also been obtained [15]. In addition, a number of naturally occurring benzoquinones such as Embelin (2,5-dihydroxy-3-undecyl-1,4-benzoquinone) have recently been shown to have potent anti-inflammatory and antioxidant activity [16]. Several studies support the efficacy of synthetic 1,4-naphthoquinone derivatives as P2X7R blockers. Thus, it was shown that aryl derivatives of 1,4-naphthoquinone (AN-03 and AN-04) had a greater potential as inhibitors of the P2X7R than standard Brilliant Blue G (BBG) and A74003 blockers [17]. These compounds effectively inhibited YO-PRO-1 dye uptake, IL-1β release and carrageenan-induced paw edema in vivo.

In previous work [18] 15 synthetic derivatives of 1,4-NQs were screened and four most effective compounds—two acyclic thioglucosides, **U-548** and **U-557,** and their tetracyclic thioglucoside derivatives, **U-286** and **U-556** were selected. An in vitro biological activity assay indicated that the selected compounds inhibited P2X7R functions in the Neuro-2a neuronal cells, suggesting their analgesic properties. The analysis of these compounds in silico indicated their potential binding to the allosteric site of P2X7R [18].

The purpose of this work is to study the anti-inflammatory activity of compounds, **U-286** and **U-548,** on RAW 264.7 macrophage cells. In present work were studied the cytotoxic, cytoprotective, antioxidant and free radical-scavenging properties of these compounds. The ability of the selected compounds to inhibit ATP/LPS-induced Ca^2+^ influx into RAW 264.7 macrophage cells and uptake of the YO-PRO-1 dye also was investigated. In addition, the inhibition of the activity of the pro-inflammatory COX-2 enzyme, ATP/LPS-induced reactive oxygen species (ROS) production, and the release of pro-inflammatory cytokines IL-1β and TNF-α in RAW 264.7 cells were inspected.

## 2. Results

### 2.1. Cytotoxic Activity of 1,4-NQs

The cytotoxicity of the tested 1,4-naphthoquinones was assessed using an MTT assay. Compound **U-286** (IC_50_ = 46 µM) had moderate toxicity, and compound **U-548** (IC_50_ = 81 µM) showed the lowest toxicity (Figure 1).

### 2.2. 1,4-NQs Inhibit ATP-Induced Ca^2+^ Influx in RAW 264.7 Cells

P2X7Rs are non-selective ion channels permeable to calcium ions by the action of extracellular ATP. Activation of purinergic receptors leads to an increase in the intracellular concentration of calcium ions [Ca^2+^]_i_. To determine the ability of the selected 1,4-NQs to inhibit P2X7Rs, RAW 264.7 cells were loaded with a Ca^2+^ selective fluorescent probe Fluo-8 AM. The binding signals of the calcium entering the cell with the probe were recorded at 37 ℃. After the addition of low concentration of ATP to macrophages primed with LPS, a significant increase in [Ca^2+^]_i_ was observed. That influx was inhibited by the P2X7R blockers, BBG (10 μM) and A438079 (10 μM) (Figure 2 a,b). 

Both 1,4-NQs at concentration range of 0.1–5.0 µM demonstrated significant inhibition of calcium influx, showing efficacy comparable to the standard blockers A438079 and BBG. Compound **U-286** caused the greatest effect, significantly reducing the influx of calcium ions at concentrations of 5.0 μM, 1.0 μM and 0.1 μM by 51.7 ± 3.8%, 54.4 ± 2.6% and 46.7 ± 6%, respectively. The treatment of cells with **U-548** also led to the inhibition of ATP-induced calcium entry at all concentrations studied (5.0 µM—32.9 ± 10.1%; 0.1 µM—46.1 ± 4.6% and 1.0 µM—43.5 ± 5.7%) (Figure 2).

### 2.3. 1,4-NQs Inhibit ATP-Induced YO-PRO-1 Dye Uptake in RAW 264.7 Cells

It is known that the stimulation of P2X7R by high concentrations of ATP leads to the opening of a macropore, capable of the uptake of large-charged molecules up to 900 Da [6]. To assess the ability of the studied 1,4-NQs to block the formation of a macropore, the low molecular weight fluorescent dye YO-PRO-1 was chosen. The treatment RAW 264.7 cells by 2 mM ATP increases the dye uptake to (208.3 ± 7.9%) relative to the control without ATP. The effect of the quinone derivatives on dye uptake was evaluated in comparison with BBG (10 μM) and A438079 (10 μM). Standard blockers BBG and A438079 significantly inhibited dye uptake by 48.3 ± 8.3% and 33.4 ± 6.7%, respectively. **U-548** exhibited the highest level of inhibition at a concentration of 5.0 μM (44.2 ± 2.5%), demonstrating a dose-dependent effect at two other concentrations (1.0 μM, 22.4 ± 1.1% and 0.1 μM, 10.7 ± 3.7%). The quinone **U-286** also affected dye uptake at the tested concentrations: 0.1 μM inhibition by 13.6 ± 1.2%; 1.0 μM—by 19.6 ± 6.5%; 5.0 μM—by 13.1 ± 2.3% (Figure 3). Thus, the selected 1,4-NQs **U-286** and **U-548** inhibit the ATP-induced formation of the P2X7 receptor pore. 

### 2.4. 1,4-NQs Protect RAW 264.7 Cells against Toxic Effect of ATP 

Since LPS is known to increase the production of inflammatory mediators and modulate the functioning of P2X7Rs in macrophages, including the RAW 264.7 line, we pre-primed the cells with LPS to enhance the action of ATP [19]. The effect of the studied 1,4-NQs on cell viability at high concentrations of ATP (2 mM) primed with LPS was evaluated using three different methods investigating various parameters of cell viability. BBG (10 μM), A438079 (10 μM) and the enzyme apyrase, reducing ATP content in extracellular media, were used as antagonists to eliminate the cytotoxic effect of ATP on cells (Figure 4).

The effect of ATP and LPS was expressed in an increase in the level of lactate dehydrogenase (LDH) in the extracellular space by 87.1 ± 8.6%, relative to the control. Compound **U-286** had a significant inhibition of the release of LDH at concentrations of 0.1 and 5.0 μM, showing effective inhibition by 43.5 ± 8.7% and 38.2 ± 3.4%, respectively. The treatment of quinone **U-548** cells at a concentration of 5.0 μM resulted in a 44.2 ± 3.3% decrease of LDH levels, while the other two concentrations (1.0 and 0.1 μM) had no significant effect (Figure 4a).

The MTT test also showed that the selected compounds increased cell viability by influencing their metabolic activity under the action of ATP and LPS. The most effective compound was **U-286**, which increased cell viability at concentrations of 0.1, 1.0 and 5.0 μM by 4.9 ± 3.1%, 9.5 ± 2.2% and 17.3 ± 3.1%, respectively. A significant increase in cell viability by 6.8 ± 0.6% was also demonstrated by compound **U-548** at a concentration of 1.0 μM (Figure 4b).

Viability analysis using fluorescein diacetate (FDA) approach leaded to a similar result. Compounds **U-286** and **U-548** at a concentration of 1.0 μM increased cell viability to the level of control without ATP and LPS (Figure 4c).

### 2.5. 1,4-NQs Inhibit ATP-Induced ROS Production in RAW 264.7 Cells

Previously, we proved that the studied substances are able to inhibit the production of ROS in neuronal cells under the action of high concentrations of ATP [18]. In the present study, the ability of compounds **U-286** and **U-548** to inhibit ROS production in RAW 264.7 macrophage cells under the action of low concentrations of ATP (300 µM) with pre-priming of cells with LPS (100 ng) was studied (Figure 5a). This ATP/LPS effect on macrophages showed an increase in ROS production by 40.2%. At the same time, treatment with only LPS or only ATP did not show a significant increase in ROS production, which may indicate the synergistic participation of these two compounds in the induction of inflammation and oxidative stress. The inhibitory effect of the studied compounds was evaluated in comparison with the action of the standard P2X7 receptor blocker A438079 at a concentration of 10 µM, significantly decreasing intracellular ROS amount by 69.7 ± 3.2% compare to ATP/LPS. As a result, it was shown that pretreatment of cells with **U-286** and at concentrations of 5.0, 1.0, and 0.1 µM demonstrated a statistically significant reduction of the ROS level by 27.5 ± 9.5%, 36.2 ± 3.8%, and 61.8 ± 6.6%, respectively. The use of compound **U-548** at a concentration of 5.0 µM did not lead to inhibition; however, concentrations of 1 and 0.1 µM significantly inhibited ROS production by 39.3 ± 4.4% and 48.6 ± 3.7%, correspondingly (Figure 5a).

### 2.6. 1,4-NQs Inhibit of the Iron-Induced Oxidation of the Mouse Brain Homogenate

It is known that 1,4-NQs may cause oxidative stress in exposed cells [20] and, therefore, have an effect on redox signaling. The antioxidant activity of synthetic 1,4-NQs and the well-known antioxidant ionol (2,6-di-*tert-*butyl-4-methylphenol) was studied using the model of non-enzymatic Fe^2+^-induced oxidation of the mouse brain homogenate (Figure 5b). Antioxidant activity was evaluated by a fluorescent method for determining of the thiobarbituric acid reactive substances (TBARS), the end products of oxidation reacting with thiobarbituric acid, in the formation of which all possible forms of active oxygen are involved. Preincubation of a mouse brain homogenate with selected compounds **U-548** and **U-286** led to a decrease TBARS content after Fe^2+^-induced oxidation of the mouse brain homogenate. Thus, the compound **U-548** led to a dose-dependent decrease in the level of ROS and TBARS content at concentrations of 0.05–10.0 μM from 13.8% till 38.4%. **U-286,** decreased in the content of TBARS, showed the greatest antioxidant effect, demonstrating dose-dependent efficacy over the entire range of studied concentrations of 0.05–10.0 µM from 36.1% to 64.0%. This antioxidant activity of the studied 1,4-NQs was comparable lower than that of ionol selected as an antioxidant control (Figure 5b).

### 2.7. 1,4-NQs Do Not Bind Diphenylpicrylhydrazyl (DPPH) Free Radicals

1,4-NQs **U-286** and **U-548** were tested for their ability to scavenge DPPH free radicals in a cell-free assay. In this investigation, the selected compounds did not show significant binding of diphenylpicrylhydrazyl (DPPH) free radicals. The maximum binding of free radicals (47.41%) was shown for **U-548** at a concentration of 100 µM. The standard antioxidants, quercetin (IC_50_ = 9.38 µM) and ascorbic acid (IC_50_ = 41.75 µM), were used as positive controls (Table 1).

### 2.8. 1,4-NQs Inhibit the Production of Pro-Inflammatory Cytokines and COX-2 Activity in RAW 264.7 Cells

According to the obtained data, the action of LPS increased the level of pro-inflammatory cytokines, TNF-α and IL-1β, in RAW 264.7 cells (Figure 6a,c). This effect is apparently due to inflammatory processes that can be induced by LPS binding to membrane Toll-like receptors. It was found that compounds **U-286** and **U-548** at concentrations of 0.1–1.0 μM were able to significantly inhibit TNF-α production in RAW 264.7 cells. The compound **U-548** showed the highest activity, which reduced the level of TNF-α at concentrations of 0.1 and 1.0 μM by 50.2 ± 9.9% and 67.9 ± 11.0%, respectively. Compound **U-286** also showed a significant inhibitory effect. At concentrations 1.0 µM and 0.1 µM this compound decreased TNF-α production by 35.6 ± 8.8% and 42.5 ± 7.7%, correspondingly (Figure 6a).

In addition, the effect of selected 1,4-NQs on the activity of mouse cyclooxygenase-2 (COX-2) in RAW 264.7 cells were studied under the action of LPS. In this experiment, both tested compounds showed a decrease in enzyme activity at concentrations of 0.1–1.0 μM. Compound **U-286** was significantly effective at both concentrations, inhibiting COX-2 activity by 82.5 ± 2.6% and 71.3 ± 1.0%, respectively (Figure 6b). Compound **U-548** showed a high inhibitory effect at a concentration of 1.0 μM by 80.3 ± 0.6%, and significantly less at a concentration of 0.1 μM by 29.4 ± 1.3% (Figure 6b).

The influence of the studied compounds on the expression of IL-1β was assessed using a Western blot analysis. Macrophage activation stimulates the release of IL-1β in a two-step process. The first step is stimulation of the toll-like receptor 4 (TLR4) by LPS, which resulted in accumulation of cytoplasmic pro-IL-1β. The second step is the ATP-dependent stimulation of P2X7Rs, promoting nucleotide-binding, leucine-rich repeat, pyrin domain containing 3 (NLRP3) inflammasome-mediated caspase-1 activation and secretion of mature IL-1β [21]. In this experiment, it was confirmed that the effect of a small concentration of LPS (100 ng/mL) had a minor effect on the release of the mature cytokine form. Therefore, a combination of a high concentration of ATP and priming by a relatively low concentration of LPS was chosen. This cell treatment resulted in a significant increase in IL-1β production, which were inhibited by 1,4-NQs. As a result, the highest level of inhibition of IL-1β expression was achieved when treated with compound **U-286** (0.1 μM) and A438079 (10 μM), while the use of compound **U-548** and BBG had no visible effect on the expression of mature IL-1 β (Figure 6c–d). 

## 3. Discussion

The P2X7R is a non-selective ion channel that is activated by exposure to ATP. Dysfunction or overexpression of the P2X7R causes numerous pathological conditions associated with pain, inflammation and tissue degeneration, such as neuropathic pain, CNS disorder, impaired hormonal response, rheumatoid arthritis, postischemic conditions, neurodegenerative and oncological diseases [7,22,23,24,25,26,27,28]. Cascades of downstream reactions triggered by P2X7R activation lead to the assembly, maturation and release of IL-1β [29], the production of the pro-inflammatory cytokine TNF-α [30], and reactive nitrogen and oxygen species formation [31]. In addition, P2X7R is directly involved in adaptive immunity by triggering the activation of T cells [32]. As a result, this receptor subtype is a very promising pharmacological target, and modulation of P2X7R exhibits potential interest in the treatment of oncological and autoimmune disorders. Thus, the usage of antagonists that will inhibit the activity of P2X7 receptors and reduce their expression levels may be effective in the treatment of associated diseases.

The selective blockers have been found for P2X7Rs such as A438079, OxATP, specific antibodies and some others [8,33]. It has been shown that these antagonists can significantly inhibit the activity of these receptors, reduce their expression level and can be used to treat concomitant diseases associated with inflammation and pain [34]. However, many of these compounds are screened out at the stage of clinical trials, do not pass blood–brain barrier, cause unwanted side effects, or have little efficacy in human trials [35].

It is known that 1,4-naphthoquinone derivatives can be used as a new class of P2X7R antagonists with high pharmaceutical potential. For example, it was shown that derivatives of 3-halo- and 3-aryl-2-hydroxy-1,4-naphthoquinone (AN-03 and AN-04) are able to inhibit the functions of the P2X7 receptor in various cell cultures and exhibit anti-inflammatory activity in vivo [17]. Then, three derivatives of 2-amino-3-aryl-1,4-naphthoquinone (AD-4CN, AD-4Me and AD-4F) were synthesized and studied, of which the compound AD-4F was found to be the most effective [36]. AD-4F inhibited dye uptake in mouse and human P2X7Rs shows the greatest activity in inhibiting human P2X7R. In addition, the AD-4F compound showed an inhibitory effect in ATP-induced IL-1β release greater than that of the standard BBG blocker, and also contributed to the relief of edema caused by carrageenan in vivo. The in silico analysis performed for the above derivatives of 1,4-naphthoquinone revealed the binding site of these compounds with the human P2X7R model in the allosteric site located in the region of the receptor pore [17,36].

In previous work, we studied four compounds: two pairs of acyclic thioglucosides, **U-548** and **U-557**, and their tetracyclic derivatives **U-286** and **U-556** [18]. The study showed that all compounds exhibit a pronounced inhibitory effect. In the course of that work, it was found that these compounds are able to decrease the ATP-induced production of ROS and NO in Neuro-2a cells, demonstrating a comparable or superior inhibition effect in comparison with BBG and PPADS. Compounds **U-286** and **U-548** significantly increased the viability of neuronal cells under the toxic effect of ATP, and also inhibited the YO-PRO-1 and EtBr dyes uptake and Ca^2+^ influx in Neuro-2a cells. Molecular docking *in silico* indicated a potential binding site for **U-286** to mouse P2X7R at the allosteric site of the receptor [18].

Relying on the previous study, compounds **U-286** and **U-548** were recognized as the low cytotoxic and most active antagonists of mouse P2X7R functions and therefore they were selected for the study of anti-inflammatory activity on macrophage cells in vitro. The RAW 264.7 macrophages were chosen as a model cell lime, which is a well-established biological model for the study of inflammation. Previously, it was shown that RAW 264.7 cells are able to express all types of P2X receptors including P2X7Rs, which makes them a suitable object for studying the activity of purinergic receptors in vitro [37]. Early studies confirm that stimulation of P2X7 receptors in this cell line results in cation channel opening, nonspecific pore formation, cytokine secretion, elimination of intracellular bacteria and cytolysis [38].

In this study, the concentration ranges of cytotoxic activity of the studied 1,4-NQs and IC_50_ values were established. However, compound **U-548** even slightly increased cell proliferation at a very low concentration range of 0.75–3.12 μM. This made it possible to select non-cytotoxic concentrations for further studies. Then, it was showed that the selected compounds significantly inhibited the functions of P2X receptors by affecting the ATP-induced (300 μM) entry of calcium ions in macrophages primed by bacterial LPS. It is well known that the response of macrophage cells to LPS is associated with the launch of a cascade of pro-inflammatory reactions through Toll-like receptors (mainly TLR4), which can stimulate additional ATP release and enhance the P2X7R response [37]. Such stimulation, in turn, increases the percentage of cell death that has undergone treatment. The level of inhibition of both compounds was comparable to the standard P2X7 receptor blockers BBG and A438079.

It is known that prolonged exposure to high concentrations of ATP on the P2X7R leads to the opening of a substantial macropore, which also causes depolarization and blebbing of cell membrane and irreversible apoptotic or necrotic changes in the cell [39]. YO-PRO-1 dye uptake was used in study to confirm the antagonistic effect of 1,4-NQs against mouse P2X7 receptors. As a result, the use of compound **U-548** achieved an inhibitory effect comparable to the BBG and A438079 at lower maximum concentrations, demonstrating a dose-dependent effect. Compound **U-286** also blocked dye uptake, but had a much smaller effect. This may indicate that these 1,4-NQs block the formation of P2X7R pore responsible for low molecular weight substances transport. The effect of these compounds on the functioning of P2X7 receptors was confirmed in tests of Ca^2+^ influx, dye uptake and effect on cell viability under the action of ATP. The results of the experiments indicate that the studied compounds are able to block the formation of the macropore of the P2X7R under the action of millimolar concentrations of ATP.

Early studies suggest that ATP significantly enhances the effect of bacterial LPS on macrophage and monocyte activation by increasing the production of inflammatory mediators such as NO and ROS in addition to numerous cytokines such as IL-1β and TNF-α. Studies have shown that the expression of pro-inflammatory cytokine genes is associated with the DNA-binding activity of the transcription factor NF-κB and suppression of the production of its IκBα isoform. The effect on these factors can be explained by the cross activity of LPS-recognizing Toll-like receptors and the influence of purinergic P2X7 receptors [40]. The level of ROS is one of the indicators of the redox status in cells. High concentrations of ATP lead to oxidative stress and an increase in this indicator through the activation of P2X7R accompanying the inflammation process. Inhibition of ROS levels by 1,4-NQs may indicate a direct effect of these compounds on these purinergic receptors and a potential blocking of inflammation processes in tissues. Thus, in the present study, it was demonstrated that the combined effect of LPS (100 ng/mL) and a relatively low concentration of ATP (300 μM) led to a significant increase in ROS production (by 40% compared to control), which was largely inhibited by A438079. The studied 1,4-NQs inhibited the production of ROS in RAW 264.7 cells, apparently suppressing the activity of the P2X7 receptor, thereby protecting cells from the damaging effects of free radicals. 

Some 1,4-NQs are known to have significant antioxidant activity [20]. Although the scavenging activity of the tested compounds was not detected in the DPPH test, **U-286** and **U-558** demonstrated an ability to reduce the content of end products of peroxidation in the mouse brain homogenate. This suggests that these compounds have an antioxidant effect and also can directly reduce the level of ROS in cells under the action of ATP, thereby exhibiting anti-inflammatory activity.

Compounds **U-286** and **U-548** also significantly increased the viability of RAW 264.7 cells in the presence of cytotoxic concentration of ATP (2 mM) upon preliminary priming of cells with LPS. To emphasize the key role of the P2X7 receptor in cell death, the enzyme apyrase was used in this experiment, which completely neutralized the effects of ATP on cells. Thus, the level of viability of cells treated with apyrase or A438079 corresponded to the level of viability in the control. The ability of the studied 1,4-naphthoquinones to protect macrophages high ATP concentration indicates that these compounds can block the P2X7R pore formation leading to cell death.

It is known that activation of P2X7Rs in macrophages, monocytes and microglia stimulates post-translational processing and release of the mature form of the cytokine IL-1β due to the efflux of intracellular K^+^ and subsequent activation of caspase-1 [41]. In performed experiments, only compound **U-286** noticeably inhibited the ATP/LPS-stimulated release of the mature form of IL-1β, similarly to the P2X7R blocker A438079, which confirms the P2X7R involvement in IL-1β production and its inhibition by 1,4-naphthoquinone. Surprisingly, compound **U-548** had no detectable effect on this process as well as BBG.

The activity of the COX-2 and the production of prostaglandin E2 (PGE2) induced by it is a direct consequence of downstream reactions caused by the release of the pro-inflammatory cytokine IL-1β in macrophages and microglial cells upstream signaling, including P2X7R. [42]. Therefore, inhibition of COX-2 activity may be the result of blocking P2X7R and the corresponding synthesis of pro-inflammatory cytokines, on the one hand, and direct inhibition of enzyme activity by 1,4-NQs, on the other hand. During the macrophage, long-term exposure to LPS compound **U-286** significantly reduced the activity of the COX-2 in RAW 264.7 cells, while **U-548** was less active. This may be due to the weak inhibitory activity (or lack thereof) of **U-548** on IL-1β production.

It is known that activation of the P2X7 receptor due to the entry of Ca^2+^ also leads to the launch of the p38/MAPK pathway and the release of the pro-inflammatory cytokine TNF-α [43]. In the present work, it was approved that compounds **U-286** and **U-548** significantly inhibit the release of TNF-α upon long-term exposure to LPS on RAW 264.7 cells, which confirms the hypothesis of their potential anti-inflammatory activity.

It was previously found that studied 1,4-NQs exhibit a weak dose-dependence in cell level tests probably because of the nonlinear manner of P2X7R inhibition. In some tests, compound **U-286** and **U-548** showed an absents of clear dose–response dependence or even reverse dependence in tests with Ca^2+^ influx, LDH release, cell viability, ROS formation, TNF-α production and COX-2 activity (Figure 2a; Figure 4a,c; Figure 5a; Figure 6a,b). Perhaps, acyclic thioglucoside, **U-548** and tetracyclic thioglucoside derivative, **U-286,** are capable to modulate P2X7R activity and at high concentrations act as partial agonists, while low concentrations block the functioning of the receptor. A similar absence of dose–response dependence was found with the same compounds in the previous study on Neuro-2a cells and P2X7Rs [18].

Based on the data, obtained studied synthetic 1,4-NQs, **U-286** and **U-548**, can be considered as effective inhibitors of the mouse P2X7R and as potential scaffolds for the development of new anti-inflammatory and analgesic drugs. In further studies, we plan to evaluate the degree of selectivity of these compounds for purinergic receptors using transgenic cell lines that stably express P2X7R. Our future research will also explore the biological activity of these compounds in models of pain and inflammation in vivo.

## 4. Conclusions

The study revealed that synthetic 1,4-naphthoquinone derivatives are capable to inhibite the pro-inflammatory functions of P2X7 receptors in macrophage cells. Compounds **U-286** and **U-548** significantly inhibited ATP-induced calcium influx and ATP-induced YO-PRO-1 dye uptake in RAW 264.7 cells. They also reduce ATP-induced ROS production, exhibit antioxidant activity and increase cell viability under the toxic effects of ATP. In the framework of this study, it was proved that these compounds have a pronounced anti-inflammatory activity in vitro, affecting the action of the pro-inflammatory COX-2 enzyme and reducing the production of the pro-inflammatory cytokine TNF-α, caused by the activation of P2X7R. The Western blot analysis detected the ability of these compounds to influence ATP/LPS-induced maturation and release of IL-1β, and reduce the level of this cytokine production in macrophage cells. The results of this study highlight these compounds as potential P2X7R blockers having anti-inflammatory properties. This data suggest these 1,4-NQs as a potential pharmacological scaffold for the development of new anti-inflammatory drugs. 

In this work, it was shown for the first time that synthetic 1,4-naphthoquinones (the chemical name of 2x naphthoquinones) suppress inflammatory processes in macrophage cells and can be considered as a basis for the creation of new targeted NSAIDs drugs, the action of which is based on blocking P2X7R.

Despite the fact that we obtained reliable results on the anti-inflammatory activity of the studied compounds in vitro, these data cannot directly indicate the anti-inflammatory activity in models of inflammation in vivo. Until now, there is no information about the pharmacokinetics of these compounds in animals and humans, as well as the side effects they cause. 

The next study will be focused on investigating and confirming the analgesic and anti-inflammatory activity of selected compounds in vivo. To this end, pain and inflammation models such as the «hot plate» test, the «acetic writhing» test, the «carrageenan inflammation» model and the murine induced arthritis model will be applied. In addition, acute, chronic and cumulative toxicity of these compounds will be investigated.

## 5. Materials and Methods

### 5.1. Synthesis of Thioglucoside ***U-548*** and It’s Tetracycle Derivative ***U-286***

Synthesis of 1,4-naphthoquinone thioglucoside derivatives **U-286** and **U-548** was carried out according to our previously published method [18] (Figure 1).

2-Chloro-3-methoxy-1,4-naphthoquinone (**1**) 111 mg (0.50 mmol) was dissolved in acetone (10 mL). To the solution, β-D-thioglucopyranose sodium salt (**2**) 110 mg (0.50 mmol) and MeOH (10 mL) were added. The reaction mixture was stirred at room temperature for 1.5 h to full conversion of initial chloroquinone (**1**). The formed orange precipitate of **U-286** was filtered, washed successively with water and acetone, then dried. The filtrate was evaporated and subjected to preparative TLC (benzene-ethylacetate-methanol, 2:1:1 *v/v*) giving 2-(β-D-glucopyranosyl-1-thio)-3-methoxynaphthalene-1,4-dione (**3, U-548**). 

2-(β-D-glucopyranosyl-1-thio)-3-methoxynaphthalene-1,4-dione (**3, U-548**).Yield 108 mg (56%), brown solid, mp 96–99 °C. R*_f_* 0.56 (benzene-ethylacetate-methanol, 2:1:1 *v/v*, silufol plates) [14]. ^1^H NMR (700 MHz, DMSO-d_6_): δ 3.09 (m, 3H), 3.22 (m, 1H), 3.34 (m, 1H), 3.53 (m, 1H), 4.11 (s, 3H), 4.33 (t, 1H, *J* 5.7 Hz), 4.93 (d, 1H, *J* 4.4 Hz), 5.10 (d, 1H, *J* 4.0 Hz), 5.29 (d, 1H, *J* 9.7 Hz), 5.46 (d, 1H, *J* 6.2 Hz), 7.82 (m, 2H), 7.95 (m, 1H), 7.97 (m, 1H). IR (KBr): 3435, 2923, 1660, 1591, 1555, 1441, 1385, 1334, 1254, 1215, 1142, 1075, 1046, 1020, 919 cm^−1^. 

(2R,3R,4S,4aR,12aS)-2-Hydroxymethyl-3,4,-dihydroxy-3,4,4a,12a-tetrahydro-2H-naphtho[2,3-b]pyrano[2,3-e][1,4]-oxathiine-6,11-dione (**4, U-286**). Yield 28 mg (16%), orange powder, mp 350–351 °C, R_f_ 0.67 (benzene-ethylacetate-methanol, 2:1:1 *v/v*, silufol plates) [14]. ^1^H NMR (300 MHz, DMSO-d_6_): δ 3.25–3.65 (m, 5H), 3.74 (ddd, 1H, *J* 10.0, 5.7, 1.2 Hz ), 4.77 (t, 1H, *J* 7.5 Hz), 4.99 (d, 1H, *J* 7.5 Hz), 5.43 (d, 1H, *J* 5.3 Hz), 5.71 (d, 1H, *J* 5.3 Hz), 7.78–7.86 (m, 2H), 7.91–8.01 (m, 2H). IR (KBr): 3464, 3350, 1644, 1590, 1568, 1405, 1266, 1181, 1075, 935 cm^−l^.

### 5.2. Compounds

Compounds **U-286** and **U-548** were synthesized according to the methodology described in [18]. Compounds were prepared as a stock solution in DMSO (10 mM) and then dissolved in ddH_2_O to the desired concentration.

### 5.3. Cytotoxic Activity Assay

RAW 264.7 cells (2 × 10^4^ cells/well) were seeded and incubated in 96-well plates for 24 h at 37 ℃, 5% CO_2_. 1,4-NQs were added to cells at final concentrations of 0.35–100.0 μM, using double dilutions. Plates were incubated for an additional 24 h. After incubation, the medium with tested compounds was replaced with 100 μL of pure medium. Cell viability was determined using the MTT (3-(4,5-dimethylthiazol-2-yl)-2,5-diphenyltetrazolium bromide) method, according to the manufacturer’s instructions (Sigma-Aldrich, Saint-Louis, MO, USA). For this purpose, 10 μL of MTT stock solution (5 mg/mL) was added to each well and the microplate was incubated for 4 h at 37 ℃. After that, 100 μL of SDS-HCl (1 g SDS/10 mL dH_2_O/17 μL 6 N HCl) were added to each well followed by incubation for 4–18 h. The absorbance of the converted dye formazan was measured using PHERAstar FS plate reader (BMG Labtech, Ortenberg, Germany) at a wavelength of 570 nm. The results were presented as percent of control data, and concentration required for 50% inhibition of cell viability (IC_50_) was calculated.

All tested compounds were added to the wells of the plates in a volume of 20 μL dissolved in PBS (DMSO concentration ≤ 1%).

### 5.4. Ca^2+^ Influx Measurement

RAW 264.7 cells were seeded in 96-well plates (4 × 10^4^ cells per well) in a DMEM cultural medium and incubated overnight at 37 ℃, 5% CO_2_. Then, the cells were washed once with HBSS saline (140 mM NaCl, 5 mM KCl, 0.8 mM MgCl_2_, 2 mM CaCl_2_, 10 mM glucose, 10 mM HEPES, pH 7.4) and loaded with 5 μM Fluo-8 AM (Abcam, Cambridge, UK) and 0.05% (*w*/*v*) Pluoronic ® F-127 (Sigma-Aldrich, Burlington, MA, USA) in the same buffer solution. Cells were incubated for 40 min (37 ℃, 5%) and then were washed with HBSS buffer without the fluorescent dye and treated by studied compounds during 20 min at RT in the dark. Standard inhibitors, non-selective P2X7R antagonist BBG (10 μM) (Sigma-Aldrich, Burlington, MA, USA) and competitive P2X7R antagonist A438079 (10 μM, Sigma, Burlington, MA, USA) were used as inhibitory controls. Fluorescence was measured with PHERAstar FS plate reader (BMG LABTECH, Ortenberg, Germany) by recording excitation signals at 490 nm and emission signals at 510 nm. ATP (300 μM final concentration) was added using a robotic microinjector after the baseline recording. 

### 5.5. YO-PRO-1 Uptake Assay

RAW 264.7 cells were seeded in 96-well plates at a density of 4 × 10^4^ cells/well and incubated in DMEM for 24 h, 37 ℃, 5% CO_2_. The cells were then washed twice with HBSS and the medium was changed to the same buffer containing YO-PRO-1 dye (Sigma, Burlington, MA, USA, 2.5 μM final concentration). Further, the studied compounds were added to the cells at concentrations of 5, 1, and 0.1 μM. Standard P2X receptor blockers BBG and A438079 at a concentration of 10 μM were used as a comparison control. The fluorescence level was measured with a PHERAstar FS plate reader (BMG Labtech, Ortenberg, Germany) at λ_ex_ = 480 nm and λ_em_ = 520 nm, ATP 2 mM (Sigma, Burlington, MA, USA) was added after baseline recording and HBSS (20 μL) was added as a negative control. 

### 5.6. Cell Line

Mouse macrophage cell line RAW 264.7 TIB-71™ was purchased from ATCC (American Type Culture Collection, Manassas, VA, USA). Cells were grown in DMEM medium (Biolot, St. Petersburg, Russia) supplemented with 10% fetal bovine serum (Biolot, St. Petersburg, Russia) and 1% penicillin/streptomycin (Biolot, St. Petersburg, Russia) in a CO_2_ incubator at 37 ℃ and 5% CO_2_.

### 5.7. Cytoprotection Activity Assay

RAW 264.7 cells (4 × 10^4^ cells/well) were seeded and incubated in 96-well plates for 24 h at 37 ℃ and 5% CO_2_ in an incubator. Compounds **U-286** and **U-548** were added to the wells at final concentrations of 5, 1, and 0.1 μM, and the plates were incubated for an additional 1 h. BBG (10 μM), A438079 (10 μM) and apyrase (30 U/mL, Sigma, Burlington, MA, USA) were also added to cells as standard blockers of ATP action. Then ATP (2 mM, final concentration) and LPS (100 ng/mL, final concentration) were added to the cells. Cells were incubated for 24 h, after which cell viability was determined using the MTT method, as described above.

### 5.8. Detection of Lactate Dehydrogenase (LDH) Release

RAW 264.7 cells (4 × 10^4^ cells per well of a 96-well plate) were seeded and incubated for 24 h at 37 ℃, 5% CO_2_ in an incubator. Then, the cells were treated with the studied 1,4-naphthoquinones at final concentrations of 5, 1 and 0.1 µM and incubated for 1 h. BBG (10 μM), A438079 (10 μM) and apyrase (30 U/mL) were used as inhibition control. Next, LPS (100 ng/mL, Sigma, St. Louis, MO, USA) was added to the cells and the cells were incubated for 4 h. Thereafter, ATP (2 mM final concentration) was added to the cells and the plates were incubated for an additional 24 h. Cells incubated without ATP and LPS were used as negative controls. The plate was then centrifuged at 250× *g* and 100 μL of the supernatant from each well was transferred to the relevant wells of an optically clean 96-well plate. An equal volume of the reaction mixture (100 μL) from the LDH Cytotoxicity Assay Kit (Abcam, Cambridge, UK) was added to each well and incubated for 30 min at room temperature. The absorbance was measured at λ = 490 nm using a Multiscan FC spectrophotometer (Thermo Scientific, Nummela, Finland). 

### 5.9. FDA Cell Viability Assay

A stock solution of fluorescein diacetate (FDA) (Sigma-Aldrich, St. Louis, MO, USA) in DMSO (1 mg/mL) was prepared. RAW 264.7 cells were seeded and treated as described above. Further, FDA solution (50 μg/mL) was added to each well and the plate was incubated in the dark at 37 ℃ for 15 min. Cells were washed and fluorescence was measured using a Fluoroscan plate reader (Thermo Labsystems, Helsinki, Finland) at λ_ex_ = 485 nm and λ_em_ = 518 nm. Cell viability is expressed as a percentage of control.

### 5.10. IL-1β Western Blotting

The expression of IL-1β was determined by Western blotting. To do this, RAW 264.7 cells were seeded in six-well plates (1.0 × 10^5^ cells/mL) and incubated in DMEM in the presence of LPS (100 ng/mL, final concentration) for 3 h. The cells were then washed twice and the cell medium was changed to PBS. Then, compounds **U-286** and **U-548** were added at concentrations of 1 and 0.1 μM, and the cells were incubated for 30 min. The standard P2X receptor blockers BBG (10 µM) and A438079 were used as inhibitory controls. After incubation with blockers, 1 mM ATP (Sigma-Aldrich, Burlington, MA, USA) was added and the cells were incubated for an additional 30 min. Cells incubated without LPS and ATP, or with only ATP, or with only LPS were used as controls. Next, the cells were washed with cold PBS (pH 7.4; BioloT, Saint Petersburg, Russia) and lysed with RIPA buffer (Sigma-Aldrich, St. Louis, MO, USA). The protein concentration in the samples was measured by the Bradford method. Proteins were separated by electrophoresis with sodium dodecyl sulfate (SDS) in 12% polyacrylamide gel. The separated proteins were then transferred to an Immobilon®-P polyvinylidene difluoride (PVDF) membrane (Merck Millipore, Darmstadt, Germany) using a semi-dry Trans-blot® SD transfer cell (Bio-Rad, Hercules, CA, USA). IL-1β protein zones were detected using specific primary polyclonal antibodies against IL-1β (P420B, Invitrogen, Waltham, MA, USA) at a dilution of 1:1000. Secondary antibodies were conjugated to horseradish peroxidase (Sigma-Aldrich, St. Louis, MO, USA). β-actin zones were detected with specific monoclonal antibodies (ab8227, Abcam, Cambridge, UK) applied at a 1:10,000 dilution and used as a loading control. The protein zones on the membranes were developed using the Pierce^TM^ ECL kit (Thermo Scientific, Rochester, NY, USA) according to the manufacturer’s instructions and visualized using the VersaDoc imaging system (Bio-Rad, Hercules, CA, USA). Protein zone densitometry was assessed using Image Lab 6.0.1 (Bio-Rad, Hercules, CA, USA).

### 5.11. Analysis of COX-2 Activity

RAW 264.7 cells were seeded in 96-well plates (2.0 × 10^4^/well) for 2 h at 37 ℃ in a 5% CO_2_ incubator. The cells were incubated 2 h after complete adhesion with **U-286** and **U-548** compounds at concentrations of 1.0 and 0.1 μM for 1 h. Then, LPS was added at a concentration of 1.0 μg/mL to each well and incubated for 24 h. Cells incubated without LPS and compounds or with LPS alone were used as positive and negative controls. To prepare a cell lysate, cells were washed once with 0.2 ml cold PBS, resuspended in 0.2 mL PBS, and centrifuged at 500× *g* for 3 min. The cell pellet was then resuspended in 0.1 mL lysis buffer with a mixture of protease inhibitors, shaken and incubated on ice for 5 min and centrifuged (12,000× *g*, 4 °C, 3 min). The collected supernatant was stored on ice for later use. COX-2 activity was determined using a cyclooxygenase activity assay kit (BN00779, Assay Genie, Dublin, Ireland) according to the manufacturer’s protocol.

### 5.12. ELISA Analysis of TNF-α Level

RAW 264.7 cells were seeded in 96-well plates (2 × 10^4^ per well) and incubated for 2 h (37 °C, 5% CO_2_) until complete adhesion. Then compounds **U-286** and **U-548** were added to the cells at concentrations of 1 and 0.1 μM. Cells were incubated with substances for 1 h, and then LPS (1.0 μg/mL) was added and incubated for 24 h. Cells incubated without LPS and compounds or with LPS alone were used as positive and negative controls, respectively. Samples were then centrifuged (1000× *g*, 20 min), and supernatants were collected and stored on ice for later use. Cells were washed gently with cold PBS and then resuspended at 1500× *g* for 10 min at 2–8°C to remove cell debris. The TNF-α level in the mixture of supernatants and cell lysates was immediately analyzed using the Mouse TNF-α ELISA kit (SEA133Mu, Cloud-Clone, Houston, TX, USA), according to the manufacturer’s instructions.

### 5.13. ROS Formation Assay

RAW 264.7 cells were seeded in 96-well plates at a density 2 × 10^4^ cells per well for 24 h. The medium was replaced with fresh DMEM and treated with LPS (100 ng/mL) for 3 h at 37 ℃. After that, the cells were treated with studied compounds (5, 1, 0.1 µM final concentrations) for an additional 30 min. Standard P2X7R inhibitor A438079 (10 µM) was used as inhibitory control. Then ATP (300 µM final concentration) was added to each well for 30 min. For fluorescence measurement, the cells were washed once and cell medium was replaced by HBSS, containing 10 µm of fluorescent dye 2,7-dichlorodihydrofluorescein diacetate (H_2_DCFDA) (Sigma, Darmstadt, Germany). Cells were incubated for 30 min at 37 ℃, washed twice with HBSS without fluorescent probe and additionally incubated for 20 min at RT in the dark. The measurement of fluorescence intensity was carried out using PHERAstar FS plate reader (BMG Labtech, Ortenberg, Germany) at λ_ex_ = 485 nm and λ_em_ = 520 nm.

### 5.14. Determination of Antioxidant Activity in Brain Homogenate

The procedure was performed according to the protocol described in [44]. The brain of male C57BL/6 mouse was washed with cold buffer (140 mM KCl, 10 mM K_2_HPO_4_, pH 7.4) and then homogenized using a Potter–Elehjem tissue homogenizer, adding the buffer to the brain in a weight ratio of 1:4. 10 mL of buffer was added to the resulting homogenate and centrifuged for 10 min at 900 rpm in a Labofuge 44R centrifuge (Thermo Scientific, Karlsruhe, Germany). The supernatant was transferred to a volumetric flask and adjusted to 25 mL with a buffer. The Lowry protein concentration in homogenate was in the range of 0.27–0.3 mg/mL.

To determine the antioxidant activity, the solution of the tested compounds was added to the brain homogenate, then 2 mM FeSO_4_ solution was added, and the reaction was stopped by adding of 20% trichloroacetic acid (TCA) (Reachim, Moscow, Russia). The homogenate was centrifuged, supernatant was collected and 0.7% solution of thiobarbituric acid (TBA) (Sigma, Saint-Louis, MO, USA) in 50% glacial acetic acid was added to reveal the products reacting with TBA. The content of TBA-reacting substanses (TBARS) was determined spectrophotometrically using PHERAstar FS plate reader (BMG Labtech, Ortenberg, Germany) by measurement of fluorescence intensity at 480/520 nm (λ_ex_/λ_em_).

### 5.15. Radical Scavenging Assay

DPPH radical scavenging activity of compounds was tested as described [45] with minor modifications. The compounds were dissolved in MeOH, and the solutions 1,4-NQs, ascorbic acid (Vekton, St. Petersburg, Russia) or quercetin (Sigma-Aldrich, Steinheim, Germany) as a positive control (120 µL) were dispensed into wells of a 96-well microplate. In all of them, 30 µL of the DPPH (Sigma-Aldrich, Steinheim, Germany) solution in MeOH (0.75 mM) was added to each well. The concentrations of compounds and ascorbic acid in mixture were 0.01–100.0 µM. The plates were incubated in the dark at room temperature for 30 min, and then the absorbance was measured at 517 nm with a Multiskan FC microplate photometer (Thermo Scientific, Waltham, MA, USA). The negative control contained no test compound. The results are presented as percentages of the negative control (DMSO) data. The final results for ascorbic acid and quercetin were reported as IC_50_, which is the concentration of compound that scavenged 50% of DPPH radicals in the reaction solution.

### 5.16. Statistical Analysis and Data Evaluation

All data were obtained in three independent replicates and calculated values were expressed as mean ± standard error of mean (SEM). One-way analysis of variance (ANOVA) with post-hoc Student–Newman–Keuls test was performed using SigmaPlot 14.0 (Systat Software Inc., San Jose, CA, USA). A significant difference between two groups was considered achieved when p was < 0.05.

## Data Availability

Not applicable.

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
