# Peer review of "Anti-Inflammatory Activity of 1,4-Naphthoquinones Blocking P2X7 Purinergic Receptors in RAW 264.7 Macrophage Cells"

_toxins, 2023, doi:10.3390/toxins15010047_

Round 1
Reviewer 1 Report
The manuscript studied the antiinflammatory effect of 1,4-naphthoquinone thioglucoside derivatives, compounds U-286 and U-548 in in RAW 264.7 cells. The authors demonstrated that U-286 and U-548 decreased COX-2 enzyme, pro-inflammatory cytokines TNF-α and IL-1β in RAW 264.7 cells, and significantly protects macrophage cells against the toxic effects of ATP and LPS.
My comments
-All catalog numbers for all kits and reagents should be added
-Why was IL-1beta measured by western blot and TNF Alpha by ELISA?
-Statistical analyses should be explained. How ere significance among groups calculated? what was the post hoc test used?
-Although the authros studied the antioxidant effect using mice brain tissue, this was not mentioned in the abstract nor the title
-Please add reference for the method used for determination of antioxidant activity in brain homogenate, why the authors used brain tissue?, why male mice were only used?
Author Response
Thank you very much for your comments and recommendations. All of them are very valuable and useful to us.
-All catalog numbers for all kits and reagents should be added
- Thank you for your notice! All missing catalog numbers were checked and added in Materials and Methods, paragraph 5.10., 5.11., 5.12.
-Why was IL-1beta measured by western blot and TNF Alpha by ELISA?
- The ELISA is more accurate and sensitive method than western blotting and therefore was chosen to detect the pro-inflammatory cytokine TNF-α and evaluate the activity of the COX-2 enzyme. However, despite the lower sensitivity, western blotting was used to screen for IL-1β, as this method provides additional information about the molecular weight of the protein. IL-1β is produced as a 31 kDa precursor (pro-IL-1β) which is further processed by caspase-1 into the mature active form of 17 kDa. Western blotting makes it possible to distinguish between mature and unprocessed IL-1β molecules and indirectly provides information on cytokine activation
-Statistical analyses should be explained. How were significance among groups calculated? what was the post hoc test used?
- Statistical analysis method was added and described in Section 5.16. This section was renamed from «5.16. Data evaluation» to «5.16. Statistical analysis and data evaluation». All data were obtained in three independent replicates and calculated values were expressed as mean ± standard error of mean (SEM). One-way analysis of variance (ANOVA) with post-hoc Student–Newman–Keuls test was performed using SigmaPlot 14.0 (Systat Software Inc., San Jose, CA, USA). Significant difference between two groups considered achieved when P was < 0.05.
-Although the authros studied the antioxidant effect using mice brain tissue, this was not mentioned in the abstract nor the title
- Abstract was changed and studied antioxidant effect was mentioned in this section.
-Please add reference for the method used for determination of antioxidant activity in brain homogenate, why the authors used brain tissue? Why male mice were only used?
- Reference to the publication in which this method was used was added in section «15. Determination of antioxidant activity in brain homogenate».
- Mouse brain tissue is composed primarily of neurons and microglia. By obtaining a homogenate from this tissue, we recreate the metabolic environment of mouse neuronal and macrophage cells used in our experiments, and the oxidative processes in them. Mouse brain tissue homogenate was also chosen due to the fact that this substrate allows one to study the antioxidant activity of compounds of biphilic origin.
- The choice of male mice in this study allows us to standardize the experiment and avoid differences in brain tissue metabolism associated with changes in the hormonal level in the brain of females, which depended on the menstrual cycle.
Reviewer 2 Report
It is interesting to read the manuscript "Anti-inflammatory activity of 1,4-naphthoquinones blocking P2X7 purinergic receptors in RAW 264.7 macrophage cells". I appreciate the authors' efforts in conducting this research, which will be more helpful to the scholars working in the area. However, it may be taken into consideration for publication in the Toxins after the following changes are made and they are included suggested points in the manuscript.
1. Abstract: I can clearly see the author's overall study's background, objective, and results. There are no methods, conclusions, or perspectives. The approach chosen to conduct this research would have been to be highlighted, also the conclusion and with suggestions for the future directions. so that after reading the abstract, readers would understand the authors' whole research.
2. Avoid “I/We/Our” throughout the manuscript. Instead use “The present/current/previous study”.
3. Include a short paragraph in the introduction discussing about inflammation, the NSAIDs that are currently on the market, and their drawbacks. Why is finding new drugs vital in this field, then? (Refer this article https://doi.org/10.3390/molecules27030734).
4. Additionally, the introduction's references should be updated with the most recent citations, such as those regarding natural benzoquinone and its anti-inflammatory properties https://doi.org/10.5958/0974-360X.2020.00618.6 .
5. Overall, the results were sufficiently detailed and well written. I appreciate the authors' efforts. However, another thing I observed was the lack of explanation provided in the discussion of each section. I would suggest the authors to refocus their discussion on each section so that it is clear how the findings from the research fit into the wider context of what is going on right now about anti-inflammatory drug discovery and developments, rather than providing more background information about the literature.
6. In the conclusion, author should provide a critical justification of the research findings. The author should increase the novelty (in the conclusion section). The importance of the research should be emphasised by the author.
7. Line 395-397, In the methodology, the author must elaborate the procedure for synthesis of U-286 and U-548.
8. The procedure in section 5.9 and 5.14 is too long. Authors should think about making it concise. even, and if possible, other procedures. Because everything is excessively lengthy and utilizes established and proven methods.
9. For the data collected during this investigation, the authors conducted a statistical analysis. However, I was unable to find the section of the statistical analysis section that specified the method used in this study and its statistically significant range.
10. Change ED50 to IC50.
Author Response
Thank you very much for your comments and recommendations. All of them are very valuable and useful to us.
- Abstract: I can clearly see the author's overall study's background, objective, and results. There are no methods, conclusions, or perspectives. The approach chosen to conduct this research would have been to be highlighted, also the conclusion and with suggestions for the future directions. So that after reading the abstract, readers would understand the authors' whole research.
- Thank you for your appreciation and very helpful comments on the article. As per your recommendation, methods, conclusions and perspectives have been added to the abstract. Our proposals for further investigation of these promising compounds were added to the final part of the manuscript.
- Avoid “I/We/Our” throughout the manuscript. Instead use “The present/current/previous study”.
- Thank you for your notice! All “I/We/Our” throughout the manuscript were deleted.
- Include a short paragraph in the introduction discussing about inflammation, the NSAIDs that are currently on the market, and their drawbacks. Why is finding new drugs vital in this field, then? (Refer this article https://doi.org/10.3390/molecules27030734).
- New short paragraph in the introduction discussing about inflammation, the NSAIDs that are currently on the market, and their drawbacks was included.
- Additionally, the introduction's references should be updated with the most recent citations, such as those regarding natural benzoquinone and its anti-inflammatory properties https://doi.org/10.5958/0974-360X.2020.00618.6
- Thanks for your recommendation. A related proposal and reference to a publication regarding the anti-inflammatory activity of natural benzoquinones has been added to the Introduction section.
- Overall, the results were sufficiently detailed and well written. I appreciate the authors' efforts. However, another thing I observed was the lack of explanation provided in the discussion of each section. I would suggest the authors to refocus their discussion on each section so that it is clear how the findings from the research fit into the wider context of what is going on right now about anti-inflammatory drug discovery and developments, rather than providing more background information about the literature.
- We refocused discussion in each section so that each results of experiments were explaned in context of general aim of this study.
- In the conclusion, author should provide a critical justification of the research findings. The author should increase the novelty (in the conclusion section). The importance of the research should be emphasised by the author.
- Paragraph on critical justification of the research findings, novelty and importance of research has been added to the Conclusion section.
- Line 395-397, In the methodology, the author must elaborate the procedure for synthesis of U-286 and U-548.
- Section 2.1 (Synthesis of 1,4-NQs) was removed from Results and added in Section 5.1. «Synthesis of thioglucoside U-548 and it’s tetracycle derivative U-286». Synthesis protocol was fully described.
- The procedure in section 5.9 and 5.14 is too long. Authors should think about making it concise. Even, and if possible, other procedures. Because everything is excessively lengthy and utilizes established and proven methods.
- Section 5.9 has been shortened, sections 5.14 and 5.15 have been merged and substantially shortened.
- For the data collected during this investigation, the authors conducted a statistical analysis. However, I was unable to find the section of the statistical analysis section that specified the method used in this study and its statistically significant range.
- The method used for statistical analysis was added and described in Section 5.16. «Statistical analysis and data evaluation»
- Change ED50to IC50.
- Your remark has been noted, ED50 was changed to IC50.
Reviewer 3 Report
The authors test the cytotoxic, antioxidant, anti-inflammatory effect of the naphthoquinone thioglucoside derivates U-286 and U-548 in RAW cells. The study is interesting and deals with the therapeutic need to find P2X7R selective blockers.
My main concerns are the following:
1. For most of the results, the compounds do not show a dose-response effect, for instance concerning the calcium uptake or LDH release in LPS/ATP treated cells or the TNF-a secretion and COX-2 activity inhibition. The authors should include a convincing explanation for this observation.
2. The authors state that LPS pretreatment increases the expression of P2X7R. If the authors have some data concerning this statement in RAW cells it should be included. In the reference cited, the increased expression of P2X7R was observed in lung parenchyma in a mouse model.
3. Do the authors have any explanation for the increased viability observed with 1 micromolar concentrations of the compounds?
4. Figure 6: error bars are missing.
5. Figure 6: How do the authors explain that the compounds increase IL-1B secretion (except with U-286 at 0.1 micromolar concentration), and also BBG? Western blot images do not have high quality. Moreover, images of triplicates should be desirable (maybe as supplementary files).
6. Section 2.1. should appear only in the methods section (5.1).
7. Figure 1d. Product U-548 at low concentration shows a high variability in cell toxicity, showing clearly cell proliferation in some experiments. An increase in the number of experiments could have decreased this variability.
Author Response
Thank you very much for your comments and recommendations. All of them are very valuable and useful to us.
The authors test the cytotoxic, antioxidant, anti-inflammatory effect of the naphthoquinone thioglucoside derivates U-286 and U-548 in RAW cells. The study is interesting and deals with the therapeutic need to find P2X7R selective blockers.
My main concerns are the following:
- For most of the results, the compounds do not show a dose-response effect, for instance concerning the calcium uptake or LDH release in LPS/ATP treated cells or the TNF-a secretion and COX-2 activity inhibition. The authors should include a convincing explanation for this observation.
- We included in the Discussion a convincing explanation for this observation: «It was previously found that studied 1,4-NQs exhibit a weak dose-dependence in cell level tests…»
- The authors state that LPS pretreatment increases the expression of P2X7R. If the authors have some data concerning this statement in RAW cells it should be included. In the reference cited, the increased expression of P2X7R was observed in lung parenchyma in a mouse model.
- Thank you for your comment. Indeed, no effect of LPS on the expression of P2X7Rs in RAW 264.7 macrophages has been shown. We removed those phrases in sections 2.2; 2.4; 2.8. Instead, a short paragraph was inserted in Section 2.4 with the corresponding reference: «Since LPS is known to increase the production of inflammatory mediators and modulate the functioning of P2X7Rs in macrophages, including the RAW 264.7 line, we pre-primed the cells with LPS to enhance the action of ATP [20].». Also the additional paragraph on LPS application was included in Section 2.8 « Macrophage activation stimulates the release…».
- Do the authors have any explanation for the increased viability observed with 1 micromolar concentrations of the compounds?
- We explain this by the absence of a linear dose-dependence of the action of 1,4-naphthoquinones. The relevant paragraph has already been inserted into the Discussion section: «It was previously found that studied 1,4-NQs exhibit a weak dose-dependence in cell level tests…».
- Figure 6: error bars are missing.
- Error bars were added.
- Figure 6: How do the authors explain that the compounds increase IL-1B secretion (except with U-286 at 0.1 micromolar concentration), and also BBG? Western blot images do not have high quality. Moreover, images of triplicates should be desirable (maybe as supplementary files).
- The increase in IL-1β secretion under the influence of ATP/LPS in the presence of BBG and the studied 1,4-naphthoquinones (except U-286 at 0.1 micromolar concentration) is statistically insignificant. The absence of a blocking action in 1,4-naphthoquinones may be due to the fact that we did not find an effective concentration. BBG is not a selective P2X7R blocker and probably did not significantly affect the functioning of this receptor under these experimental conditions. The size of the Figure 6d has been enlarged and the image quality of the Western blot has been improved. This is a representative image. The original images of Western blots have already been sent to the Editors.
- Section 2.1. should appear only in the methods section (5.1).
- Section 2.1 (Synthesis of 1,4-NQs) was removed from Results and added in Section 5.1. in Materials and Methods.
- Figure 1d. Product U-548 at low concentration shows a high variability in cell toxicity, showing clearly cell proliferation in some experiments. An increase in the number of experiments could have decreased this variability.
- Yes, indeed, U-548 at low concentrations increased cell proliferation. The relevant sentence has been inserted in the Discussion section: «However, compound U-548 even slightly increased cell proliferation it at very low concentration range of 0.75 – 3.12 μM…».
Round 2
Reviewer 1 Report
The authors addressed my comments and the manuscript has been significantly improved
Reviewer 2 Report
The author responded to all of my inquiries and revised the manuscript content accordingly. As a result, I suggest that it be taken into consideration for publication in Toxins in its present form.